# Porphyrin as Diagnostic and Therapeutic Agent

**DOI:** 10.3390/molecules24142669

**Published:** 2019-07-23

**Authors:** Ncediwe Tsolekile, Simphiwe Nelana, Oluwatobi Samuel Oluwafemi

**Affiliations:** 1Department of Chemical Sciences (formerly Applied Chemistry), University of Johannesburg, P. O. Box 17011, Doornfontein 2028, Johannesburg, South Africa; 2Department of Applied Chemistry, University of Johannesburg, P. O. Box 17011, Doornfontein 2028, Johannesburg, South Africa; 3Department of Chemistry, Cape Peninsula University of Technology, P.O. Box 652, Cape Town 2000, South Africa; 4Department of Chemistry, Vaal University of Technology, Private Bag X021, Vanderbijlpark 1900, South Africa

**Keywords:** porphyrin, conjugate, PTT, PDT, MRI, theranostic, dual modality

## Abstract

The synthesis and application of porphyrins has seen a huge shift towards research in porphyrin bio-molecular based systems in the past decade. The preferential localization of porphyrins in tumors, as well as their ability to generate reactive singlet oxygen and low dark toxicities has resulted in their use in therapeutic applications such as photodynamic therapy. However, their inherent lack of bio-distribution due to water insolubility has shifted research into porphyrin-nanomaterial conjugated systems to address this challenge. This has broadened their bio-applications, viz. bio-sensors, fluorescence tracking, in vivo magnetic resonance imaging (MRI), and positron emission tomography (PET)/CT imaging to photo-immuno-therapy just to highlight a few. This paper reviews the unique theranostic role of porphyrins in disease diagnosis and therapy. The review highlights porphyrin conjugated systems and their applications. The review ends by bringing current challenges and future perspectives of porphyrin based conjugated systems and their respective applications into light.

## 1. Introduction

The early detection of diseases plays a vital role in their successful treatment and recovery. However, currently available detection methods such as biopsy procedures (i.e., endoscopic procedures, lumbar puncture, bronchoscopy, and pelvic examination) [1], blood tests (i.e., tumor marker tests and circulating tumor cell tests) [2], and diagnostic tests (i.e., CT and MRI scans) [3] are highly costly, time-consuming, with complex operations that require skilled personnel to operate. Following these tedious diagnosis processes, the selection of appropriate treatment often follows, and in some cases like cancer, requires the use of multi-treatment modalities. In an effort to address some diagnostic problems and subsequent delays in treatment, researchers and clinicians have looked into the development of economically viable and easily synthesizable materials that are able to provide both a diagnostic and therapeutic effect within the listed diagnostic tools. Porphyrins are a unique class of compounds that are widely available in nature. They have distinctive photo-physical properties and are highly stable with a wide absorption profile which spans from the ultraviolet (UV) region to near-infrared (NIR) region (Figure 1a) [4]. Their excitation results in electron movement into an excited state which is followed by either fluorescence, phosphorescence, or intersystem crossing into an excited triplet state (fully explained by the Jablonski diagram) [5]. The ability of the porphyrins to release fluorescence enables their use as diagnostic tools and in fluorescence-guided tumor dissections and imaging. Their ability to undergo inter-system crossing into excited triplet state (which results in the production of singlet oxygen) (Figure 1b) allows for their use in therapeutic applications such as photodynamic and photo-thermal therapy [6,7]. Surface modifications of porphyrins have enabled the control their physico-chemical and pharmacological properties, thus permitting their use in a variety of other applications. Biological application of porphyrins is inundated with reports on their use in photodynamic therapy and magnetic resonance imaging. Progressive research on porphyrins within these fields has resulted in great scientific and industrial interest in porphyrins, metalloporphyrins, and their related compounds. The bio-application of porphyrins has thus expanded into bio-sensors, fluorescence tracking, in vivo MRI, and PET/CT imaging to photo-immuno-therapy [8,9]. Porphyrins have shown great potential within the medical imaging field (or techniques that create visual representation of the interior of the body for clinical and medical intervention) due to their pharmacological properties such as low toxicity, high tumor uptake, and the possibility of forming complexes with metals. A good imaging agent ideally is non-destructive and allows for imaging with minimal background signal from cellular auto-fluorescence, properties that are associated with recently synthesized porphyrins or porphyrin conjugated systems [10]. It is worth noting that porphyrin analogues, such as phthalocyanines and chlorines, have attracted a lot of attention and have been the subject of several interesting reviews [11,12]. Although there has been an emergence in the application and studies of porphyrin analogues, their research has tended to highlight them as a separate discipline within the generalized porphyrins. Metalloporphyrins (porphyrins with metal ions within their inner cavity) have also been found to be very useful (Figure 1c). The inclusion of metal ions often enhances the properties of the free base porphyrin and introducing properties (i.e., improved phototherapeutic efficacy, providing MRI contrast agent, and enhanced Raman imaging just to mention a few) that would otherwise not have been observed in the porphyrin alone [13]. Moreover, the development in the application of metalloporphyrin has shown them to confer great theranostic potential. Herein, this review focuses on recent developments on the use of porphyrins (including their conjugation systems) from a diagnostic and therapeutic perspective. It highlights the therapeutic role of porphyrins and porphyrin-conjugated systems in photo-medicine techniques such as photodynamic and photo-thermal therapy. The review analyzes the diagnostic role of porphyrin in medicinal imaging techniques, such as photoacoustic, magnetic resonance, and fluorescence imaging. It further highlights the dual functionality of porphyrins conjugated systems and reviews some of the available examples of these materials. 

## 2. Therapeutic Role of Porphyrins

An ideal drug should be able to deliver the appropriate drug quantities at a controlled rate and time to the desired site. However, a number of challenges in this respect have been reported with current drug and/or delivery systems suffering from altered bio-distribution, clearance issues from the body, and uncontrolled release with non-matching pharmacodynamics [16]. Research has thus focused on the development of targeted drug delivery systems to address the above mentioned short-comings and porphyrins have to date contributed significantly in this research arena. The preferential localization of porphyrins in tumors and their ability to generate reactive singlet oxygen and low dark toxicities [17] have resulted in their use in various biological applications. However, their inherent lack of bio-distribution due to water insolubility has shifted research into porphyrin-nanomaterial conjugated systems to address this challenge. Porphyrins have a number of structural features that make them highly suitable for alterations, subsequent conjugation, and application. For example, the structure of porphyrin consists of four pyrrole units that are interconnected via methine bridges; therefore, the presence of the N-H groups allows for the binding of anions through hydrogen-bonding and the presence of nitrogen atoms within the core structure of porphyrin further allows for selective metal ion chelating, and polymers and liposomes conjugation just to mention a few [18,19,20]. These structural features have therefore shifted research towards porphyrin biomolecules, whereby their respective bio-distribution, aggregation, stability, and toxicology is being extensively examined. A number of porphyrin post-functionalization strategies focusing on both core-functionalization (whereby the core-skeleton of the porphyrin is directly functionalized at the β- and/or meso- position) and the peripheral functionalization (where the functional groups are introduced on the peripheral substituent) have since been reported. Although a number of functionalization strategies have been reported, literature has mainly focused on porphyrins bearing metal cores, halogens, and alkynes. [21]. Table 1 provides summarized data on the conjugation of porphyrins with various moieties. The design and construct of porphyrins with multiple functionalities has made them useful for bio-imaging, cancer targeting, and cancer therapy. As seen in Table 1, conjugation of porphyrins has become a state-of-the-art approach for efficient improvement of applications relating to porphyrins for diagnostic and therapeutic application.

### 2.1. Porphyrins in Photo-Medicine

The tunability of porphyrins to absorb in the therapeutic window (600–800 nm) has allowed for their use in photo-medicine applications such as photodynamic therapy (PDT) and photo-thermal therapy (PTT). The use of porphyrins within these therapies is a result of their inherent ability to generate singlet oxygen. The generation of singlet oxygen by porphyrins results from their excitation in ground state by light of a specific wavelength into excited singlet state. The excited porphyrin undergoes intersystem crossing to the triplet excited state (T^1^) and reacts with molecular oxygen in the surrounding environment to form singlet state oxygen (^1^O_2_). In photodynamic therapy, porphyrins based photosensitizers are the most widely used as photosensitizing drugs. PDT porphyrin based photosensitizers (PBP) are categorized into three generations. First-generation PBP have seen limited use due to long half-life, chemical impurity, low attenuation coefficients, and excessive accumulation onto the skin. The second-generation PBP seemingly addressed the short-comings of the first generation; however, they lacked the desired water solubility required for biological application and the ability of the required light to penetrate into tissue for the treatment of deep-seated tumors. This has resulted in the development of third-generation PBP with increased cancer cell-specific accumulation. Third-generation PBP are centered on the use of conjugated systems of the porphyrins linked to moieties such as targeting bio-molecules, nanomaterials (e.g., QDs such CuInS/ZnS and Au), and other moieties (i.e., SPIONS) to improve the specificity of the porphyrins [22,23,24,25,26].

#### Porphyrin—Nano-Conjugates

The use of superparamagnetic iron oxide nanoparticles (SPION)—porphyrin conjugates has allowed for directional light-application during PDT due to the ability of SPIONs to attract cancer cells via the magnification properties of the SPION. Our group [27] reported on the use of methoxypolyethyleneglycol-thiol-SPIONs-gold-meso-tetrakis(4-hydroxyphenyl)porphyrin conjugate (nano-drug) as magnetic field enhancer for PDT. The results showed that the nano-drug exhibited high photo-toxicity against the MCF-7 breast cancer cells in the presence of light (10 J/cm^2^ at 673 nm) after exposure for 14 min 51 s. However, when the cells were exposed to an external magnetic field, higher cellular uptake and greater photo-toxicity than the non-exposed cells were observed, suggesting that magnetic targeted PDT is a better treatment modality than non-magnetic field-driven PDT. Feng et al., 2016 [22], reported on the use of 5-aminolevulinic acid (ALA) conjugated to ternary quantum dots (CuInS/ZnS) for Forster resonance energy transfer (FRET) mediated PDT under NIR femtosecond laser. The group successfully performed PDT using 800 nm and 1300 nm femtosecond lasers to achieve the destruction of the cancer cells. In a recent development, we also reported [14] improved singlet oxygen quantum yield generation (0.27 for mTHPP and 0.72 for the conjugate) using the same CuInS/ZnS QDs conjugated to 5, 10, 15, 20-meso(4-hydroxyphenyl) porphyrin (mTHPP) with high water solubility and stability for over six months. These reports are just a few of many that have shown improvements achieved with photo-physical and bio-applicability of porphyrin conjugated systems.

## 3. Porphyrins in Medicinal Imaging

Porphyrins have shown their applicability in multidisciplinary fields (Figure 2) [28]. With the development of imaging devices, they have emerged as promising imaging tools for diagnostic and therapeutic monitoring purposes. Due to their low cytotoxicity in the absence of light, tunable photo-physical properties, such as absorption and emission wavelength, superficial derivatization, and superior tumor uptake [29,30], they have been used in bio-imaging applications such as photoacoustic imaging (PAI), magnetic resonance imaging (MRI) [7], x-ray radiography [31], and photoacoustic [32] and fluorescence imaging [33].

### 3.1. Magnetic Resonance Imaging (MRI)

Magnetic resonance imaging (MRI) has been available since the early 1970s as a sensitive, non-invasive imaging diagnostic tool. It enables 3D imaging of the human body in the absence of ionization radiation. In MRI, porphyrins are used as contrast agents. An ideal contrast agent is one that is able to affect radiofrequency pulses and enhance image contrast on relaxation time weighted MR image (properties that are present in co-ordinated metal ions) [11,34]. This has led to the advancement in research of porphyrins chelated with metal ions such as Gd [35], Mn [11], and Cu [36] to form metalloporphyrins as contrast agents. Metalated meso-sulphonatophenyl porphyrin (TPPS_4_) (Figure 3) is one of the first macrocyclic porphyrins to be used as a contrast agent in MRI. Improved porphyrin based contrast agents have since been reported, such as the use of C60-manganese porphyrin [37] gadolinium±diethylenetriamine pentaacetic acid [38] and Mn- meso-sulphonatophenylporphyrin [39]. The combination of 1,1-dicyano-2-[6-(dimethylamino) naphthalene-2-yl] propene (DDNP) carboxyl derivative to functionalize the surface of SPIONs was used by Zhou et al., 2014 [40], as a contrast agent in MRI for Alzheimer’s disease imaging. The DDNP-SPIONs were analyzed for their MRI relaxation properties using MR imaging and the study demonstrated high T_2_ relaxivity of 140.57 s^−1^ FemM^−1^. An in vitro experiment of the DDNP-SPIONs binding to β-amyloid aggregates showed that the combination (DDNP-SPIONs) induced the fluorescence enhancement of the DDNP-SPIONs. The research by Venter et al., 2018 [9], on gadolinium free Mn porphyrin amine as a MRI contrast agent for T1-based cellular imaging and tracking of human embryonic has shown great potential of porphyrins as contrast agents as their study reported on the oncological capabilities of chelated porphyrins.

### 3.2. Photoacoustic Imaging (PAI)

Photoacoustic imaging (PAI) is emerging as a promising imaging technique. It uses exogenous imaging agents to further increase the contrast and specificity of imaging or target specific molecular processes. Guided by tumor-targeting nano-platforms, PAI application is used to accurately locate tumors and has found use in clinical applications such as oncology, rheumatology, and in the cardiovascular field [41]. Clinical application of PAI requires exogenous imaging agents/contrast agents with high absorption coefficients in the optical window to enable tissue imaging [42]. The use of porphyrins conjugated systems has thus been found to be very useful in this regard. Banala et al., 2017 [32], reported on the use of quinone-fused porphyrins as contrast agents for PAI application. They used black porphyrins due to their absorption which is in the PAI suitable NIR region (706–733 nm). The group applied PAI using 0.31 mm (i.d.) polypropylene tubes in a water chamber phantom. The study reported on the metal effect insertion on the PAI properties and found that inserting Zn(II) in porphyrins generated the strongest signal (3.2 times higher intensity) than that for indocyanine green (ICG), a commonly used PAI agent. Recently, Wu et al., 2018 [43], reported on the use of porphyrin-implanted carbon nano-dots for photoacoustic imaging, photodynamic therapy, and in-vivo breast cancer ablation. The porphyrin-implanted carbon nano-dots were prepared by partial and selective pyrolysis of 5,10,15,20-tetrakis(4-aminophenyl) porphyrin (TAPP) and citric acid (CA) to form porphyrin-implanted carbon nano-dots (PNDS). Cetuximab was used to functionalize the PNDS to form C225-PNDS. The PNDS showed near-infrared absorption with deep tissue penetration and spatial resolution as contrast agent for photoacoustic molecular imaging with deep tissue penetration. The excellent photodynamic therapeutic effect and the enhanced photoacoustic contrast ability of C225-PNDs were validated in mice bearing MDA-MB-231 breast cancer cells. The results obtained in this study showed the new class of porphyrin-implanted carbon nano-dots as promising material for numerous and expanding biological applications.

### 3.3. Fluorescence Imaging

Fluorescence imaging involves the visualization of the area of interest using fluorescent materials (i.e., porphyrins) as labels for molecular processes or structures. It is widely used for experimental observations including the location and dynamics of gene expression, protein expression, and molecular interactions in cells and tissues. However, the use of fluorescence imaging suffers a great deal of photo-bleaching. The use of Zn-metalated porphyrins functionalized with symmetrical two phenylethynyl groups as fluorescence imaging guided cancer photodynamic therapy provides evidence of the prestige applicability of porphyrins in fluorescence imaging. Cationic porphyrins have found use in positron emission tomography (PET) for their water solubility and strong DNA binding capabilities. Furthermore, [68Ga]-labeled cationic porphyrin, viz. 5,10,15,20-tetra(4-methylpyridyl) porphyrin (TMP), has found use in PET radiotracer for tumor imaging. Preliminary bio-evaluation studies on [68Ga]-labeled TMP showed potential as radio-labeled porphyrin derivative for tumor imaging [44].

Battisti et al., 2017 [33], reported for the first time red fluorescence of *Helicpbat pylori* (Hp) biofilms using protoporphyrin IX (PP IX) and co-porphyrin I (PI) for fluorescence labeling of bacteria (Figure 4). In their research, Battisti’s group used lab adapted bacterial strains, Hp ATCC 43504 and a virulent strain (cagA+ and vacA+), Hp ATCC 700824. Bacterial porphyrins were extracted from Hp. Following the design and development of LED based robotic pills, spectroscopic features of the Hp endogenous porphyrins were determined on bacterial extracts. The results showed a possible niche area in need of much research for the development of antimicrobial PDT. Photo-physical properties of porphyrins and their ability to act as functional receptors for different metal ions and formation of chelation with metal ions extend their application to metal ion sensing. Moreover, porphyrins allow for the detection of metals via electronic absorption, fluorescence, colorimetric, and electrochemical methods [4].

Cationic porphyrins are currently the most explored porphyrins due to their capability to bind to a wider range of different structural moieties (Figure 5) and their ability to dissolve in aqueous solutions due to their amphiphilic nature [46]. Bio-application of cationic porphyrins is advantaged by the amphiphilic structure of cationic porphyrins which enables the hydrophilic end to allow drug administration by making it water soluble while the hydrophobic terminal facilitates cell entry and accumulation [47]. As highlighted earlier in this review, porphyrins have found use in a number of applications, these, however, are not limited to bio-applications.

It is therefore worth mentioning the photocatalytic research that has been recently reported using porphyrin. Krieger et al., 2017 [48], reported on the use of cationic 5,10,15,20-tetrakis(4-trimethylammoniophenyl)porphyrin tetra(p-toluenesulfonate) (TAPP) and 5,10,15,20-tetrakis(1-methyl-4-pyridinio)porphyrin tetra(ptoluenesulfonate) (TMPyP) assemble with PSS-polyanions to improve on the photocatalytic activity of the oxidation of iodide to triiodide. The use of nanostructure of dendrimer–porphyrin assemblies (Figure 6) exhibited a 1.5 times higher catalytic activity for the degradation of the anionic dye methyl orange than the corresponding porphyrin only. This was attributed to the arrangement in porphyrin-J-aggregates within the assemblies. This is showing applicability of porphyrins in not only catalysis but also in sensing.

## 4. Dual-functionality of Porphyrin Conjugates

The killing of two birds with one stone concept has given rise to the now popular concept of “dual functionality” or “dual modality”. Within this concept, porphyrins bring their ability to generate singlet oxygen (Figure 7) and structural functionality (be it in the core- or peripheral functionalization) while the other moieties are brought in to improve on the bio-distribution, fluorescence, and specificity. Although porphyrins alone have shown synergetic effects on cancer cells, their conjugation to nanoparticles such as gold [49] and graphene [50] has shown that porphyrin conjugation improve their performance in dual functionality (Figure 8) treatments. Photo-thermal therapy (PTT) is emerging as a great therapeutic treatment for cancer metastasis. It operates on the principle of generating heat from near-infrared (NIR) light via photo-thermal agents (e.g., gold nanoparticles).

Compared to other conventional cancer therapies, PTT exhibits unique advantages which include temporal and spatial selectivity, high specificity to tumor, and minimal invasiveness to surrounding normal tissues [51]. The biocompatibility of gold nanoparticles and their photo-thermal features (i.e., the ability to absorb in the near-infrared (NIR) light at the plasmon resonant wavelength and then transforming light energy into heat energy) has received much attention in PTT. However, alone, Au nanoparticles are unable to generate singlet oxygen and have been shown to aggregate. Conjugating of Au NPs to porphyrins with the addition of light irradiation has proved to address the challenges of Au as PTT agents [52]. Improved antitumor efficacy in PTT has thus been reported with some studies reporting dual-modality (photodynamic and photo-thermal therapy) with hyaluronan-coated FeOOH@polypyrrole nanorods [53] and albumin/sulfonamide stabilized iron porphyrin metal organic framework nanocomposites [54].

### Dual-Imaging as Dual “Diagnostic” Functionality 

Diagnostic imaging has long been used as a tool to recognize, define, and identify tumors. However, the concept of dual/multi-imaging was only introduced in the early 2000s. Prior to that, clinicians often used trial-and-error with regards to the use of imaging as means of diagnosis. Medical imaging has since advanced with the introduction of nuclear medicine (where radioactive substances are applied for the diagnosis and treatment of disease). Lou et al., 2017 [56], chelated porphyrin-phospholipid (POP) with ^64^Cu. ^64^Cu was used to radiolabel doxorubicin (DOX) loaded liposomes. The group then applied positron emission tomography (PET) imaging to estimate the location of 4T1 mammary tumors. To complement PET, fluorescence imaging using a NIR fluorescence fiber optic probe was used to pinpoint the exact tumor site. The use of ^64^Cu-Dox-PoP liposomes in PET and fluorescence imaging to identify and quantify liposome accumulation further shows the capabilities of porphyrin conjugates within dual functionalities.

## 5. Current Challenges

There are three distinctive pathways for the synthesis of meso substituted ABCD-type porphyrins: (a) Total synthesis, (b) mixed condensation, and (c) functionalization of preformed systems [57]. Although these synthesis methods have been tried and tested, the synthesis of porphyrins bearing two different meso-substituents (Figure 9) has proved highly difficult due to scrambling and polymerization phenomena. The condensation of mixed aldehydes to prepare porphyrins has resulted in the formation of mixed porphyrins which have proved highly difficult to separate.

Self-aggregation of porphyrins still remains as one of the major challenges reported by many researchers. This is particularly experienced with anionic porphyrins such as tetrakis (4-sulfonatophenyl)-porphyrin and its derivatives which results in suppressed fluorescence and reduction in the singlet oxygen generation. A number of strategies have been reported to resolve the aggregation challenges commonly reported. However, to date, all reported studies are application based [58]. Consensus on the self-aggregation of di-acidic form of TPPS_4_ is also still not fully reached by researchers and therefore still remains a widely investigated research area. The misuse and over-prescription of antibiotics has resulted in the increase in antibiotic resistance in gram-positive and gram-negative bacteria. This has resulted in the need for alternative methods such as antimicrobial photodynamic therapy (aPDT) for disease treatment. The photo-physical properties of porphyrins allow for their use in aPDT with meso-tetraarylporphyrins as the most commonly used porphyrins in aPDT. Porphyrins such as 5,10,15-tris (1-methylpyridinium-4-yl)-20-(pentafluorophenyl) porphyrin tri-iodide [59], tetrakis (4-carboxyphenyl)porphyrin (TCPP) [60], and 5,10,15-triphenyl-20-(4-aminophenyl)porphyrin (H_2_TriMAPP) [61] and their respective conjugates are amongst the porphyrins that have been reported for aPDT application. Although some research has been reported on the antimicrobial PDT antibiotic resistance-activity using porphyrin, a number of challenges still need to be addressed, these include: (1) The in vitro inactivation of microorganisms in clinical set-up, (2) selectivity towards microbes at acceptable levels of host tissue damage in the area of interest/treatment, and (3) the efficacy of the porphyrin PSs in aPDT application in the presence of other biological matrix. Some studies have aimed at addressing these short-comings, but to date there is no clinical evidence (which would allow for aPDT as primary treatment modality) and there is no certainty that re-growth following aPDT will not occur. Moreover, there are limited reports on the efficacy of porphyrin conjugated systems on multi-resistant microbial strains [62,63,64].

## 6. Future Perspectives and Conclusion

Herein, we report on the use of porphyrins and their related compounds in a variety of diagnostic tools (i.e., MRI, CT, PAI, PET) and treatment modalities (i.e., PDT, PTT). Although significant research has been done in presenting the potential of these highly aromatic molecules as the “next generation” drugs, a lot of research still needs to be exploited as limited research has been explored in relation to their bio-distribution, mode of action/mechanism of action in-vivo, and most importantly, in their production up-scaling. The conjugation of porphyrins with various moieties ranging from small sized nanomaterials to high molecular weight polymers has also been exploited. However, limited literature is available on their pre-clinical work in different in-vivo cancer models. Despite porphyrins having proved to be highly efficient in PDT application with positive clinical trials, their application as first line cancer treatment (even in skin cancers) is still not well exploited compared to other cancer treatment modalities. Moreover, there has been no reported attempt to proceed towards clinical trials of porphyrin conjugated systems which would draw closer towards drug development of these sophisticated systems, even though the use of porphyrins in dual modality has provided a fascinating and state-of-the-art approach in disease diagnosis and treatment. Numerous porphyrin derivatives or analogues have been synthesized, such as the chlorines, phthalocyanines, and naphthalocyanines, for diagnostic and theranostic application. Furthermore, these porphyrin derivatives are known for their improved stability compared to bare porphyrins, higher coefficient extinctions (~105/M/cm) and longer wavelengths in the near-infrared region. They suffer from a number of synthetic and functionalization challenges. Moreover, there is limited literature on their toxicity profiles and improved synthetic methods aimed to improve their biocompatibility. It is therefore necessary that the porphyrin analogues are further exploited as their NIR capabilities would be highly advantageous in optical imaging techniques. To conclude this review, it is also worth mentioning pharmacological properties of porphyrins (i.e., half-life, volume of distribution, etc.) as an untouched focus area both by scientific researchers and industrial practitioners. This has further hampered the progression of porphyrin use in a clinical set-up. Moreover, the lack of reported tuning of synthetic methods to enable synthesizable porphyrin agents with dual or tri-modality functionalities has further contributed to the lack of porphyrin conjugated systems being credited as bio-compatible systems. Therefore, there is a need for exploration beyond in vitro and in vivo studies.

## Figures and Tables

**Figure 1 molecules-24-02669-f001:**
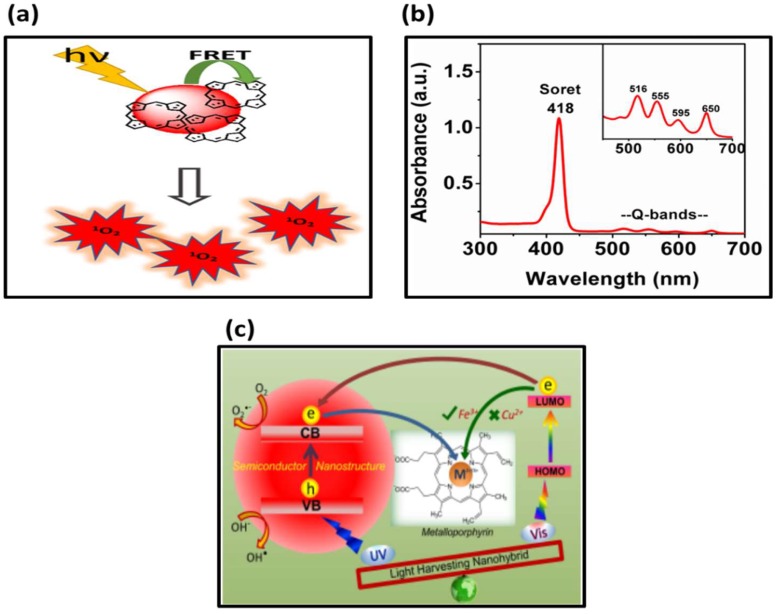
(**a**) Schematic diagram for generation of singlet oxygen [14]; (**b**) UV absorption spectra of free base porphyrin] [14]; (**c**) Metallated porphyrin functionalized with naphthalene-2-yl propenes (NPs) [15].

**Figure 2 molecules-24-02669-f002:**
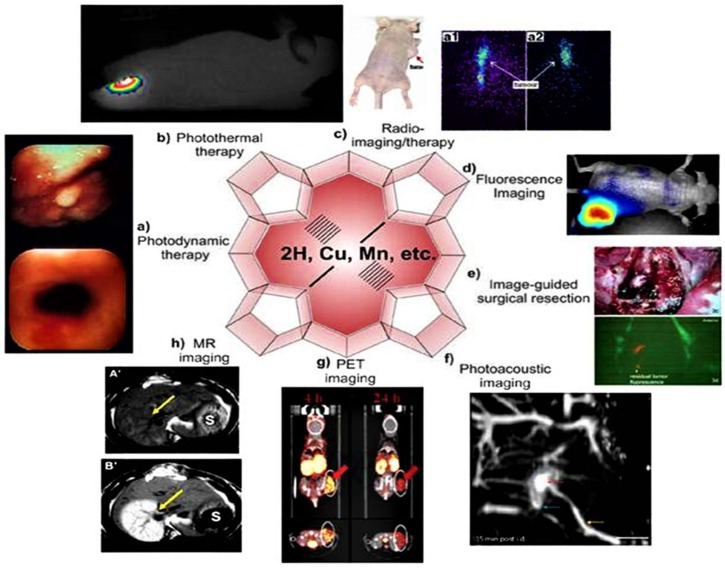
Diagram showing the multi-applications of porphyrins (**a**,**b**,**e**) including different imaging capabilities (**c**,**d**,**f–h**) of porphyrins [12].

**Figure 3 molecules-24-02669-f003:**
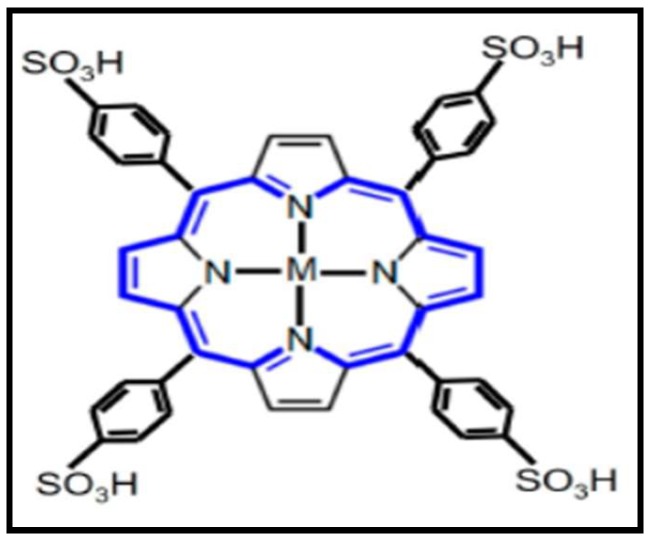
Typical structure of meso-sulphonatophenyl porphyrin which can be used as a contrast agent for magnetic resonance imaging (MRI) (M = Mn(III), Fe(III), and Cu(II) [5].

**Figure 4 molecules-24-02669-f004:**
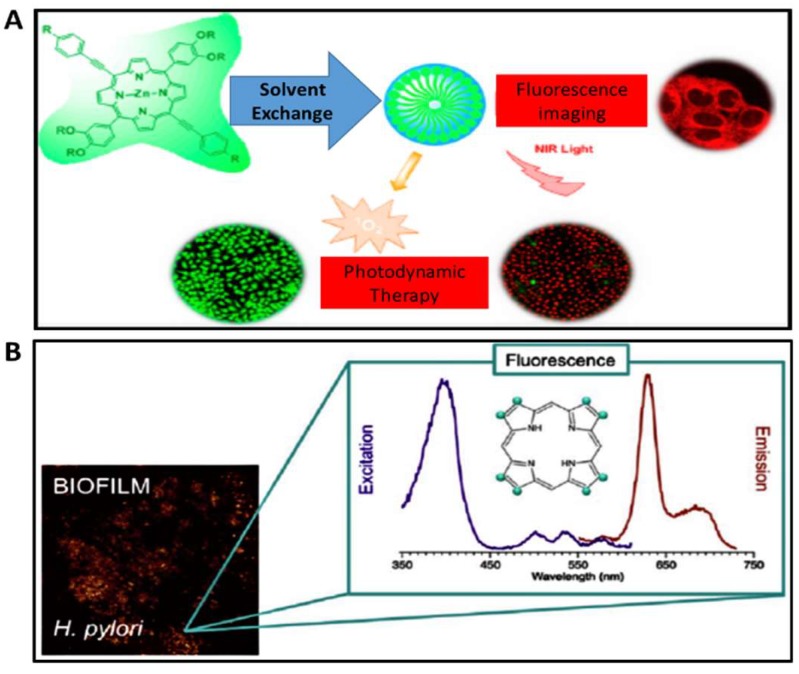
(**a**) Photodynamic therapy (PDT) application [45] and (**b**) fluorescence imaging capabilities of porphyrins [33].

**Figure 5 molecules-24-02669-f005:**
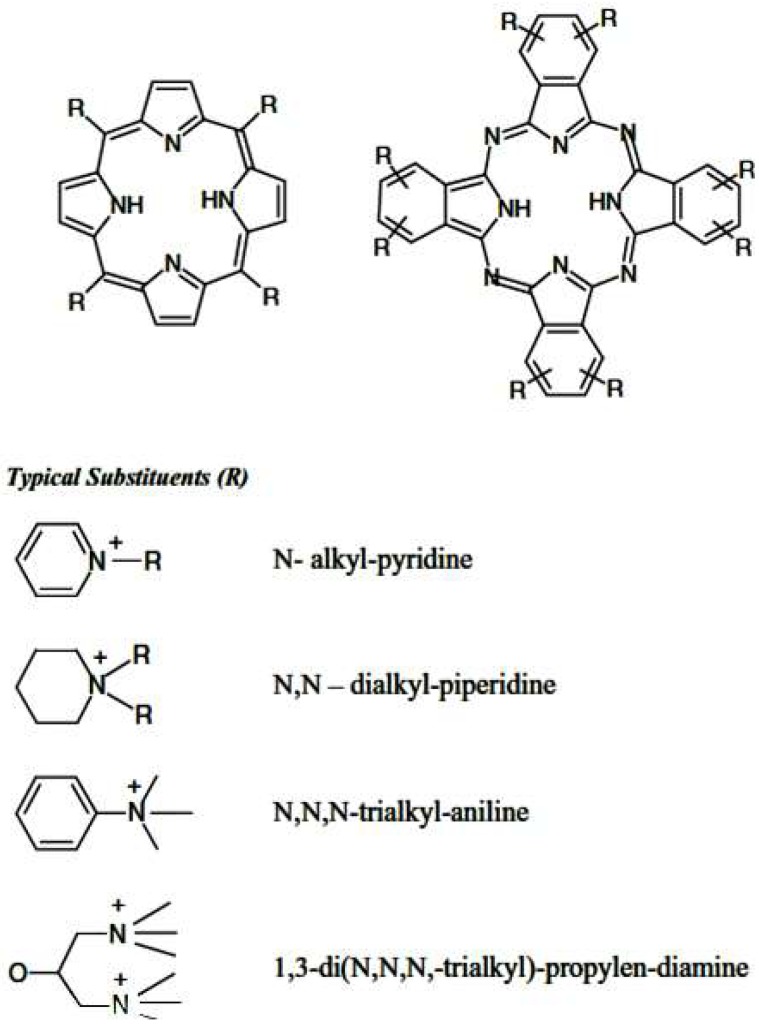
Structures of cationic porphyrin derivatives [46].

**Figure 6 molecules-24-02669-f006:**
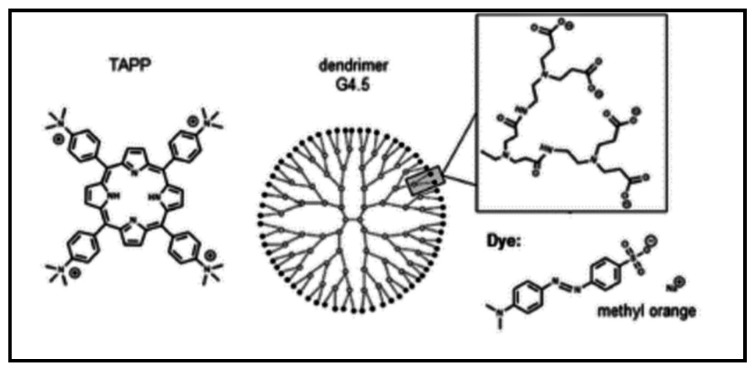
10,15,20-tetrakis(4-trimethylammoniophenyl)porphyrin tetra(p-toluenesulfonate) (TAPP). Functionalization with dendrimer [48].

**Figure 7 molecules-24-02669-f007:**
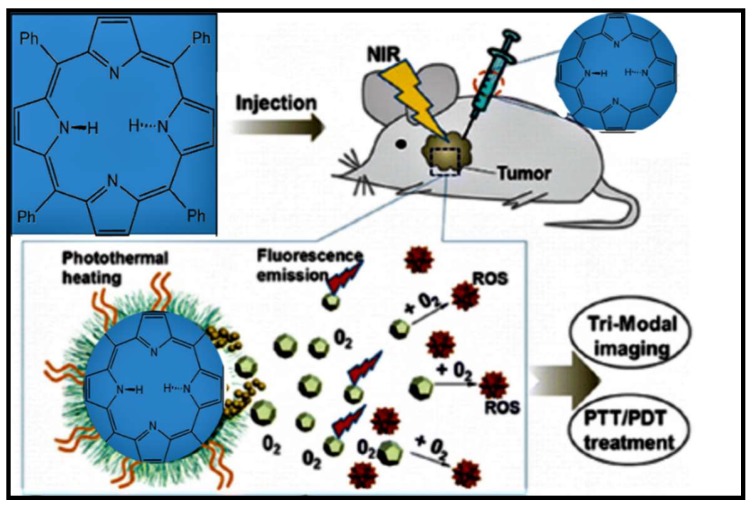
Photo-thermal therapy (PTT) and PDT treatment using singlet oxygen producing porphyrin [51].

**Figure 8 molecules-24-02669-f008:**
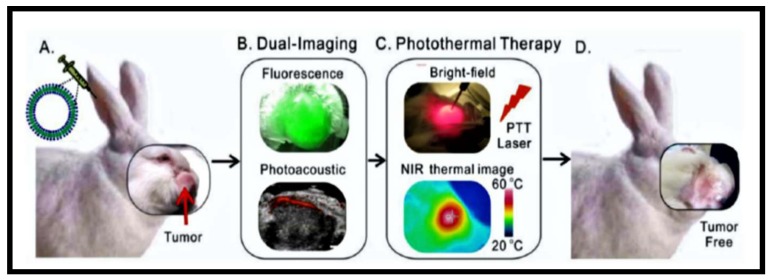
Dual functionality of porphyrins, (**A**) intravenous injection of the porphyrin, (**B**) dual (fluorescence and photoacoustic) imaging, (**C**) photothermal therapy application, and (**D**) post-treatment tumor free model [55].

**Figure 9 molecules-24-02669-f009:**
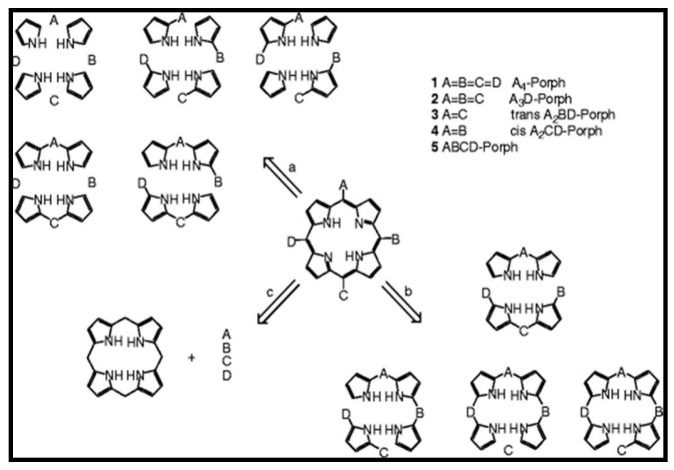
Retro-synthetic pathways to ABCD porphyrins.

**Table 1 molecules-24-02669-t001:** Porphyrin conjugated systems.

	Conjugate	Application	Reference
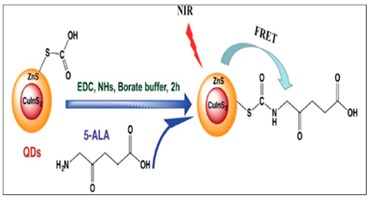	5-Aminolevulinic acid-CuInS/ZnS conjugate	Photodynamic Therapy	[22]
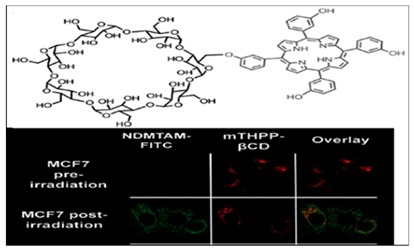	5,10,15,20-tetrakis(4-hydroxyphenyl)-porphyrin -β-cyclodextrin conjugate	Photochemical Internalization	[23]
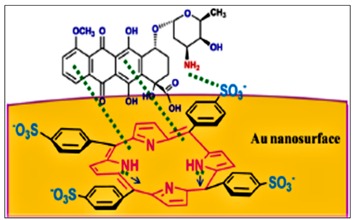	meso-tetrakis- (4-sulfonatophenyl) porphyrin-Gold conjugate	Drug delivery	[24]
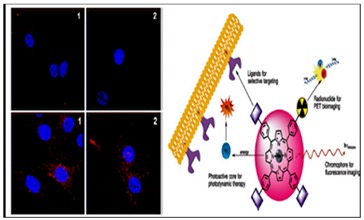	2-fluoro-2-deoxy glucose functionalized heterobifunctional glycoporphyrins	Theranostic agent (multi-modal imaging and photodynamic therapy)	[25]

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
