# Peer review of "Porphyrin as Diagnostic and Therapeutic Agent"

_molecules, 2019, doi:10.3390/molecules24142669_

Reviewer 1 Report

In  this work, Tsolekile et al. present a review porphyrins as theranostic tools. Overall, this is an interesting manuscript that provides readers a good vantage point of this topic. The figures are interesting and the text is concise. Overall, I just have a few minor comments:

1) Some of the figures appear to be "stretched" awkwardly. I recommend avoiding that phenomenon

2) The authors should mention in more detail about the PET imaging capabilities of porphyrin to directly bind 64Cu for theranostic applications like monitoring drug deliver with PET and NIRF (Luo et al., ACS Nano, 2017, 11,12482-12491). The authors should mention  that the metal chelating capabilities of porphyrin confer great theranostic potential (Shao et al., Coord Chem Rev 379, 2019, 99-120)

Author Response

1)       Some of the figures appear to be "stretched" awkwardly. I recommend avoiding that phenomenon.

Response: The figure quality has been improved.

2)       The authors should mention in more detail about the PET imaging capabilities of porphyrin to directly bind 64Cu for theranostic applications like monitoring drug deliver with PET and NIRF (Luo et al., ACS Nano, 2017, 11,12482-12491). The authors should mention that the metal chelating capabilities of porphyrin confer great theranostic potential (Shao et al., Coord Chem Rev 379, 2019, 99-120)

Response: Thank you for the interesting reads and suggestions, both articles have been included in the review (page 2 (line 64-69) and 10 (296-308).

Reviewer 2 Report

The manuscript entitled “Porphyrin as Theranostic Tool: An update on Porphyrin as Diagnostic and Therapeutic Agent” reviews the role of porphyrins in therapeutics and diagnostics.

The review is organized in the following manner:

1. Introduction

2. Porphyrin conjugated systems

3. Porphyrins in photo-medicine

3.1. Porphyrin – Nano-conjugates

4. Dual-functionality of porphyrin conjugates

5. Porphyrins in medicinal imaging

5.1. Photoacoustic imaging (PAI)

5.2. Magnetic resonance imaging (MRI)

5.3. Fluorescence imaging

6. Current challenges

7. Future perspectives and conclusion

Only one small sub topic reviews the theranostic properties of porphyrins, which is sub-topic 4. Sub topic 3 refers to porphyrins in PDT and PTT, while topic 5 deals with medical imaging.

Theranostic, is a term coined to refer to diagnostic and therapeutic effect in one single molecule. This is not the case in all topics but 4. The way to prevent that the readers may erroneously assume this article as an article in Theranostics, is to change the title accordingly, also for grammatical purposes. Perhaps it would be more suitable to have: “Porphyrins as Diagnostic and Therapeutic Agents”

I also do not understand the concept of update. Update on what previous review? Or something else? All this information should be given in the introduction of the article.

Regarding that, The introduction seems quite out of the topic, if the authors are reviewing the role of porphyrins as therapeutics and diagnostic compounds. It must be completely rewritten and put in context. As it is presented, the introduction is very naïve and misplaced.

In the introduction, the authors must explain the type of review, if it critical, extensive, or with selected examples, time frame, etc. One must find  in introduction what is going to read further.

There are many great reviews on any of the topics here reviewed. These should also be mentioned in introduction.

Most importantly: what is the criterion for choosing publications to review? Once again, is it the date, the importance, something else?

I consider the organization of the review very confuse. For instance: on point #2, the authors mention conjugation of porphyrins as very important sub-topic, I agree, but then in the next sub-topic, (Porphyrins in photo-medicine) conjugated porphyrins are also reviewed. Unless point 2 refers to conjugated porphyrins that do not fit any of the subsequent sub-topics, then it is pointless to have sub-topic 2. The therein-reviewed porphyrins should be included in the corresponding sub-topics.

In my opinion, besides the changes above, the authors should organize the review as, for instance, first mentioning the therapeutic role of porphyrins, then the imaging role, and then finally, the theranostic. This, to give purpose to the review. Otherwise, is just an amalgam of disconnected data.

Anyway, the review should be rewritten, to be again reconsidered for publication, since the actual state does not allow further refereeing. After complete rewriting, then it can be refereed properly.

Author Response

Comments and Suggestions for Authors

The manuscript entitled “Porphyrin as Theranostic Tool: An update on Porphyrin as Diagnostic and Therapeutic Agent” reviews the role of porphyrins in therapeutics and diagnostics.

The review is organized in the following manner:

1. Introduction

2. Porphyrin conjugated systems

3. Porphyrins in photo-medicine

3.1. Porphyrin – Nano-conjugates

4. Dual-functionality of porphyrin conjugates

5. Porphyrins in medicinal imaging

5.1. Photoacoustic imaging (PAI)

5.2. Magnetic resonance imaging (MRI)

5.3. Fluorescence imaging

6. Current challenges

7. Future perspectives and conclusion

Only one small sub topic reviews the theranostic properties of porphyrins, which is sub-topic 4. Sub topic 3 refers to porphyrins in PDT and PTT, while topic 5 deals with medical imaging.

Theranostic, is a term coined to refer to diagnostic and therapeutic effect in one single molecule. This is not the case in all topics but 4. The way to prevent that the readers may erroneously assume this article as an article in Theranostics, is to change the title accordingly, also for grammatical purposes. Perhaps it would be more suitable to have: “Porphyrins as Diagnostic and Therapeutic Agents”

I also do not understand the concept of update. Update on what previous review? Or something else? All this information should be given in the introduction of the article.

Response: The title has been changed as suggested by the reviewer. Thank you.

Regarding that, The introduction seems quite out of the topic, if the authors are reviewing the role of porphyrins as therapeutics and diagnostic compounds. It must be completely rewritten and put in context. As it is presented, the introduction is very naïve and misplaced.

Response: The introduction has been completely re-written.

In the introduction, the authors must explain the type of review, if it critical, extensive, or with selected examples, time frame, etc. One must find  in the introduction what is going to read further.

Response: The format/review context has been included in the introduction.

There are many great reviews on any of the topics here reviewed. These should also be mentioned in introduction.

Response: Additional references to reviews have been added.

Most importantly: what is the criterion for choosing publications to review? Once again, is it the date, the importance, something else?

I consider the organization of the review very confuse. For instance: on point #2, the authors mention conjugation of porphyrins as very important sub-topic, I agree, but then in the next sub-topic, (Porphyrins in photo-medicine) conjugated porphyrins are also reviewed. Unless point 2 refers to conjugated porphyrins that do not fit any of the subsequent sub-topics, then it is pointless to have sub-topic 2. The therein-reviewed porphyrins should be included in the corresponding sub-topics.

In my opinion, besides the changes above, the authors should organize the review as, for instance, first mentioning the therapeutic role of porphyrins, then the imaging role, and then finally, the theranostic. This, to give purpose to the review. Otherwise, is just an amalgam of disconnected data.

Anyway, the review should be rewritten, to be again reconsidered for publication, since the actual state does not allow further refereeing. After complete rewriting, then it can be refereed properly.

Response: The review has been re-organized and formatted taking into consideration all the above suggestions.

Reviewer 3 Report

This is a review on the potential medicinal applications for porphyrins. It is reasonably well written but contains a few grammatical errors and contains words that are misused or inappropriate. The review is not particularly original but brings a fresh look at this important area. The introduction is a little simplistic. For instance, porphyrins cannot simply be considered to be [18]annulenes. The structures of porphyrins and metalloprophyrins are said to "commonly" be "determined using 1H NMR and UV/visible spectroscopy". This fits well with the first chapter in Kevin Smith's Porphyrins and Metalloporphyrins, but that book was published in 1975 and things have moved on somewhat. Still, this is still essentially correct even though MS, X-ray, CV, etc. are also commonly used. The introduction contains the majority of the grammatical errors. The rest of the review introduces the numerous areas that porphyrins are being applied to in biological problems. It is worth noting that the emphasis is almost exclusively on meso-tetraaryl porphyrins and the synthetic concerns discussed on page 10 exclusively apply to that type of structures. meso-Unsubstituted porphyrins barely get a look in but that type of motif is exclusively found in nature. There is also a great deal of work being carried out on porphyrin analogues, e.g. expanded porphyrins, as well as phthalocyanines. Furthermore, chlorins have many valuable features as well. Clearly future work is likely to include many types of porphyrinoid structures in addition to the meso-substituted porphyrins. Symmetrical meso-tetrasubstituted porphyrins are easily synthesized, so that is a benefit. However, as is noted on page 10, asymmetrical structures of this type are far more challenging.

The following is a list of minor errors and suggestions for rewording:

Line 46: space after "visible"

Line 53: what do the authors mean by "inter convention" ?

Line 60: the authors write that "porphyrins is in-dated". Perhaps they mean "inundated".

Line 67: What do the authors mean by "logical blood washout"?

Line 69: Apparently "properties are embedded with recently synthesized porphyrins". Perhaps "embedded" could be replaced with "associated".

Lines 76-80: In two sentences, the authors use the word "drug" five times. Perhaps some attempt could be made to be less repetitious. 

Line 88: The pyrrole units in porphyrins are not "interconnected via methylene bridges". The connections are methine bridges. Methylene implies a CH2 unit.

Line 89: "binging of anions". I assume this should be "binding".

Line 98: "porphyrins"

Line 107: "tunability".    Line 112: "undergoes"

Line 136: "Forster"

Line 166: "have been shown"

Line 184: "in the cardiovascular"

Line 187: "regard" not "regards"

Line 210: "constrating". Presumably this should be "contrast". This is better that "contrasting".

Line 233: "i.e."

Lines 266/267: Perhaps split the sentence: "applications. However, these are not limited ..."

Line 274; I would replace "accounted" with "attributed"

Lines 275/276: The last sentence in this paragraph does not make sense.

Line 281: what does "75" refer to?

Line 283: "due to" not "due the too"

Line 284: "aldehydes"

Line 293: "reported. However, to date, all reported ..."

Line 295: I am not sure what the authors mean by "still remains a widely research area". 

Line 304: "still needing to be addressed". The rest of this sentence is muddled and needs to be rewritten.

Line 309: "short-comings, but to date there is"

Lines 319/320: "exploited. However, limited ..."

Line 331: "systems. Therefore, there is a need for exploration ..."

The use of a great deal of technical jargon makes the paper less accessible and it is made worse when the terms are incorrectly used. Nevertheless, light editing should take care of the problems.

The references are not consistently formatted. For instance, the first reference should be "J. Am. Chem. Soc.". A period should follow each abbreviation but if a part of the journal name is not an abbreviation there should be no period. Journal names should be consistently italicized. Also, reference 48 is unclear. Is this a book? No details are given.

Author Response

Comments and Suggestions for Authors

This is a review on the potential medicinal applications for porphyrins. It is reasonably well written but contains a few grammatical errors and contains words that are misused or inappropriate. The review is not particularly original but brings a fresh look at this important area. The introduction is a little simplistic. For instance, porphyrins cannot simply be considered to be [18]annulenes. The structures of porphyrins and metalloprophyrins are said to "commonly" be "determined using 1H NMR and UV/visible spectroscopy". This fits well with the first chapter in Kevin Smith's Porphyrins and Metalloporphyrins, but that book was published in 1975 and things have moved on somewhat. Still, this is still essentially correct even though MS, X-ray, CV, etc. are also commonly used. The introduction contains the majority of the grammatical errors. The rest of the review introduces the numerous areas that porphyrins are being applied to in biological problems. It is worth noting that the emphasis is almost exclusively on meso-tetraaryl porphyrins and the synthetic concerns discussed on page 10 exclusively apply to that type of structures. meso-Unsubstituted porphyrins barely get a look in but that type of motif is exclusively found in nature. There is also a great deal of work being carried out on porphyrin analogues, e.g. expanded porphyrins, as well as phthalocyanines. Furthermore, chlorins have many valuable features as well. Clearly future work is likely to include many types of c structures in addition to the meso-substituted porphyrins. Symmetrical meso-tetrasubstituted porphyrins are easily synthesized, so that is a benefit. However, as is noted on page 10, asymmetrical structures of this type are far more challenging.

Response: Thank you, noted. The introduction has been re-written taking the above points into consideration. The authors also noted the reviewer view on other porphyrin analogues and stand in agreement with the reviewer. However, the authors felt that the inclusion of porphyrin analogues (i.e. pthalocyanines and chlorines) would take away the jest of the porphyrin review as a great deal of work and reviews on these (pthalocyanines and chlorines) address them as entirely separate entities from general porphyrins (i.e. meso-tetraaryl porphyrins) which the review focuses on. Nonetheless, the future work has been revised to include phthalocyanines, chlorins and porphyrinoid structures over and above the meso-substituted porphyrins.

The following is a list of minor errors and suggestions for rewording:

Line 46: space after "visible"

Line 53: what do the authors mean by "inter convention" ?

Line 60: the authors write that "porphyrins is in-dated". Perhaps they mean "inundated".

Line 67: What do the authors mean by "logical blood washout"?

Line 69: Apparently "properties are embedded with recently synthesized porphyrins". Perhaps "embedded" could be replaced with "associated".

Lines 76-80: In two sentences, the authors use the word "drug" five times. Perhaps some attempt could be made to be less repetitious. 

Line 88: The pyrrole units in porphyrins are not "interconnected via methylene bridges". The connections are methine bridges. Methylene implies a CH2 unit.

Line 89: "binging of anions". I assume this should be "binding".

Line 98: "porphyrins"

Line 107: "tunability".    Line 112: "undergoes"

Line 136: "Forster"

Line 166: "have been shown"

Line 184: "in the cardiovascular"

Line 187: "regard" not "regards"

Line 210: "constrating". Presumably this should be "contrast". This is better that "contrasting".

Line 233: "i.e."

Lines 266/267: Perhaps split the sentence: "applications. However, these are not limited ..."

Line 274; I would replace "accounted" with "attributed"

Lines 275/276: The last sentence in this paragraph does not make sense.

Line 281: what does "75" refer to?

Line 283: "due to" not "due the too"

Line 284: "aldehydes"

Line 293: "reported. However, to date, all reported ..."

Line 295: I am not sure what the authors mean by "still remains a widely research area". 

Line 304: "still needing to be addressed". The rest of this sentence is muddled and needs to be rewritten.

Line 309: "short-comings, but to date there is"

Lines 319/320: "exploited. However, limited ..."

Line 331: "systems. Therefore, there is a need for exploration ..."

The use of a great deal of technical jargon makes the paper less accessible and it is made worse when the terms are incorrectly used. Nevertheless, light editing should take care of the problems.

Response:  The above have been corrected as suggested.

The references are not consistently formatted. For instance, the first reference should be "J. Am. Chem. Soc.". A period should follow each abbreviation but if a part of the journal name is not an abbreviation there should be no period. Journal names should be consistently italicized. Also, reference 48 is unclear. Is this a book? No details are given.

Response: Thank you, the references have been corrected. Reference 48 is a book chapter and has also been edited.

Round  2

Reviewer 2 Report

The changes are satisfactory and now the manuscript may be accepted for publication